# Experiences and Functional Health Outcomes Associated with a Walking Football Program in Rural Older Adults: A Pilot Study

**DOI:** 10.3390/sports13080272

**Published:** 2025-08-15

**Authors:** Stephen Cousins, Kylie McIntyre, Philip Lyristakis, Keanu Padula, Jane McCaig, Brett Gordon

**Affiliations:** 1La Trobe Rural Health School, La Trobe University, Bendigo, VIC 3552, Australia; k.mcintyre@latrobe.edu.au (K.M.); p.lyristakis@latrobe.edu.au (P.L.); k.padula@latrobe.edu.au (K.P.); j.mccaig@latrobe.edu.au (J.M.); b.gordon@latrobe.edu.au (B.G.); 2Holsworth Biomedical Research Centre, La Trobe University, Bendigo, VIC 3552, Australia

**Keywords:** healthy ageing, walking football, older adults, physical activity, rural health

## Abstract

Background: This pilot study aimed to investigate the experiences of participating in a brief walking football intervention among rural older adults and explore the functional health outcomes of participation. Methods: This multi-methods study saw 13 older adults (7 males/6 females, 63.2 ± 9.4 years) complete 1-h of walking football per week for six weeks. Pre- and post-intervention, participants underwent assessments of body composition, as well as functional assessments using the Senior Fitness Testing Battery. On completion of the walking football intervention, all participants joined in a semi-structured focus group interview to explore their experiences of participation. Results: Thematic analysis of focus group data identified three themes that captured participant’s experiences, including: (1) “Opportunity to jump back in with what we can physically do”, (2) Social connection and feeling “part of something bigger”, and (3) “It’s how our brain is engaged”. Trivial changes were observed in assessments of body composition (0.07–0.08) and flexibility (0.02–0.19). Furthermore, small-to-large magnitude changes were observed for several functional health outcomes suggestive of improved walking capacity (0.72–1.6), agility (−0.9) and upper and lower body muscular strength (0.49). Conclusions: Rural older adults reported experiencing perceived health and well-being improvements from participation in a brief walking football intervention, with functional health benefits also evident; however, further suitably powered evidence is highly warranted.

## 1. Introduction

Australia has an aging population [1] that is accentuated in rural areas, due to out-migration of younger individuals to metropolitan areas and an in-migration of older adults [2]. Associated with this demographic shift is an increased prevalence of chronic, degenerative health conditions [3], which can lead to decreased functional independence and quality of life [4]. Furthermore, older adults residing in rural areas experience higher rates of morbidity, lower quality of life and lower social functioning compared to older adults residing in metropolitan areas of Australia [5]. Consequently, there is a recognisable need to provide sustainable health promotion and physical activity participation opportunities for older adults living in rural areas, which can be achieved through participation in sporting pursuits. Regular and adequate participation in physical activity promotes healthy aging, realised through improvements in physical and psychological function [6].

The Person Environment Occupation (PEO) model emphasizes the complex interaction between an individual, their environment, and the activities (or occupations) they participate in [7]. This model helps to identify and address factors that influence older people’s participation in sporting pursuits, and create supportive contexts to enhance engagement, health, and psychosocial wellbeing. Physical activity participation, particularly in group/team environments, can promote social connection in older populations [8], reducing the risk of loneliness and social isolation in older adults living in rural Australia. Despite the established evidence base and guidelines for older adults to be physically active, only one-quarter of Australians aged ≥65-years continue to meet recommended physical activity guidelines [9]. Issues and concerns around health, safety, resources, knowledge, and environment are factors that inhibit older adults from being active [10], and strategies to overcome these barriers are necessary.

Adaptive sports, such as walking sports, are emerging as effective solutions to increase engagement in physical activity for older adults [11]. In adaptive sports, the rules are modified to enable walking instead of running and physical contact is usually prevented, thereby lowering the impact and risk of injury [12]. Walking football is gaining global recognition, as a safe, enjoyable, sustainable, feasible, and cost-effective approach for recruiting and retaining older populations to a physical activity program [13,14,15]. Physical, cognitive, and mental health and well-being improvements have all been suggested from participating in walking football [16,17,18]. In a review of the limited evidence-base, Corepal et al. [19] concluded that walking football provided opportunities for enjoyment; development of a team identity; and moderate-intensity activity with cardiovascular, musculoskeletal, and metabolic benefits. However, this evidence is limited by focusing on male populations residing in metropolitan areas and failing to understand how rural-residing older adults experience adaptive sports. The lack of women and inclusion of populations from rural locations has made it difficult to give generalizable recommendations regarding participation in walking football. Consequently, further research is needed to (1) identify the experiences from participating in walking football and (2) explore how walking football influences the health of both older males and females residing in rural Australia. This knowledge is needed to ensure programs are designed to meet the unique needs of rural residing older populations given they are often limited by distance and access to such programs.

Therefore, the aim of this pilot study was to investigate experiences of participating in a brief walking football intervention, along with potential improvement in implementation, in rural older adults. The secondary aim was to explore the functional health outcomes of participation.

## 2. Materials and Methods

### 2.1. Study Design

Multiple methods were used for this study. A single-arm pre-post design was used to investigate the influence of participation on functional health outcomes. A focus group interview was then conducted to explore participants’ experiences of the walking football intervention and their perspectives on the personal health and psychosocial benefits of participation. Ethics approval was obtained from the La Trobe University Human Research Ethics Committee (HEC24132, 11 April 2024). Participation was voluntary and all participants provided informed consent prior to commencement of the research.

### 2.2. Participants

Thirteen participants, seven males and six females (mean ± SD; age: 63.2 ± 9.4 years; height: 1.7 ± 0.1 m; body mass: 84.0 ± 21.1 kg; body mass index (BMI): 28.2 ± 7.0 kg/m^2^) were recruited in April and May 2024 using convenience-sampling from a walking football group in rural Victoria, as defined using the Modified Monash Model code MM2 (regional centre in, or within 20 km road distance of, a town with a population greater than 50,000) [20]. Potential participants were notified about the research via flyers distributed on physical and virtual notice boards. To be considered eligible, participants had to be aged 50-years and over with no evidence of severe movement or coordination disorders; no cognitive impairments impacting their ability to understand written and verbal English, and no contraindications to performing physical exercise.

### 2.3. Measures

Participants completed the Exercise Sport Science Australia (ESSA) Adult Pre-Exercise Screening System to determine their eligibility to participate in the study. Once deemed eligible, anthropometric data was measured in duplicate. If subsequent values differed by more than 5% a third measure was taken and the median recorded. Height was determined using a portable stadiometer (Holtain, Crymych, UK) using the stretch-stature method [21]. Body mass was determined using Seca, Model 770 (seca, Hamburg, Germany) scales. BMI was calculated from height and mass using kg/m^2^.

Assessments of functional outcomes were performed by adopting procedures described previously for the administration of the Senior Fitness Test Battery [22] comprising a 30-s chair stand test (CST), 30-s arm curl test (ACT), 8-foot timed up and go test (TUG), chair sit and reach test (CSRT), back scratch test (BST), 2-min step in place test (2MST), and 6-min walk test (6MWT). Due to participants’ limited availability both the 2MST and 6MWT were performed during the same session. Whilst this contradicts protocol recommendations, the 6MWT was performed at the start of each session and the 2MST at the end to attempt to mitigate the risk of fatigue affecting performance. Participants were informed that they could slow down or rest, if necessary, during both assessments.

The CST comprised the number of full stands that could be completed, from a seated position in 30-s with arms folded across the chest. For the ACT, the number of bicep curls that were completed in 30-s holding a hand weight of 2 kg for females and 4 kg for males was recorded. The use of these weights is inconsistent with the original recommendations, which specify 5 lb. (approximately 2.3 kg) and 8 lb. (approximately 3.6 kg) for females and males, respectively; however, these differences were unavoidable due to differences in the metric systems adopted between countries. The TUG comprised the time in seconds required to get up from a seated position, walk 8 feet as fast as possible, turn and return to the seated position. For the CSRT, participants were instructed to sit on the edge of the chair, one foot flat on the ground, the other fully extended and with both hands-on top of one another and reach as far as they could towards the toes. The distance (cm) from fingertips to toes was recorded, with a positive score recorded if participants could go past the toes, and a negative score recorded if participants could not touch the toes.

For the BST, participants were instructed to stand upright, and place one hand behind the head to reach as far as possible down the middle of the back and place the other hand behind the back and reach upwards towards the other hand attempting to touch or overlap the middle fingers of both hands. The distance (cm) between the tips of the middle fingers was recorded with a positive score recorded if participants could overlap the fingertips, and a negative score recorded if participants could not touch the fingertips. For the 2MST, participants were instructed to stand up straight next to the wall, with a level marked on the wall corresponding to midway between the patella and iliac crest. Over the course of 2-min, the participant stepped in-place until the knee reached the marked level on the wall, and the number of times the height was successfully reached was recorded. Finally, for the 6MWT, participants were instructed to walk as quickly as possible for 6-min up and down a 20-m walkway marked off in 5-m segments with the distance walked recorded and used for analysis.

### 2.4. Intervention

All assessments and games took place in a community sports facility comprising an indoor hard-court sports hall. Participants undertook baseline assessments of anthropometry and functional capacity and one week later, commenced a 6-week walking football program. Throughout the 6-weeks, participants attended a single training session each week. Sessions typically began at the same time of day (6:30 p.m.) and were 60-min in duration. Each session comprised a 10-min standardised dynamic warm up facilitated by a qualified exercise scientist, followed by a series of 3 × 12-min small-sided games, after which a 10-min dynamic and static cool-down was performed. All games took place on a surface marked by cones measuring 15 × 8 m and were refereed by a qualified volunteer walking football referee from Football Victoria. During all games no physical contact between players was permitted, running was not allowed, and participants always had to have one foot on the ground. Furthermore, the ball could not be kicked above head height, with re-starts taking place via a pass into the playing area. One week after the last walking football game, all participants underwent post-test data collection where participants completed the same tests as baseline.

### 2.5. Focus Group Interview

On conclusion of the walking football intervention, a single 60-min focus group was conducted with research participants. All participants (n = 13) were invited to participate via an online and printed flyer, although only 10 participants were available at the time of the focus group interview and volunteered to participate.

A semi-structured interview format with open-ended questions was used to explore participants’ experiences of the walking football intervention and their perspectives on the personal health and psychosocial benefits of participation. Questions included in the semi-structured interview guide included: (1) Can you tell me about your decision to participate in walking football? (2) Is this a new activity for you, or have you previously been involved in team sports? (3) How would you describe your experience of participating in walking football? What were the positive aspects? What were the negative aspects? (4) Do you have any stories about your participation in this activity? (5) Does participating in walking football change how you see and feel about yourself? (6) Would you recommend this activity to others? If so, what groups of people might benefit and why?

Inclusion of a small number of participants (n = 10) ensured there was opportunity for interaction between the participants and points of agreement, conflict, and uncertainty could be revealed [23]. Specifically, a researcher with experience in interviewing (K.M.) conducted the focus group in a private meeting room, ensuring participant confidentiality and active encouragement for all participants to share their unique perspective. Ground rules were established at the commencement of the interview to promote respectful listening and reassure participants that all viewpoints were valuable. The researcher further took an active role throughout the focus group interview, using probing and follow-up questions to deepen the conversation and explore differing opinions [24].

The focus group was audio recorded and transcribed verbatim to ensure an accurate account of the conversation was available for data analysis.

### 2.6. Statistical Analysis

Focus group interview data were analysed using thematic analysis as described by Braun and Clarke [25]. Data analysis was performed in six steps: (1) reading and re-reading the focus group content, considering content and interaction between the participants, (2) initial coding through use of line-by-line analysis, (3) identification of emerging themes, (4) review of emerging themes in relation to the entire data set, (5) identification of overarching themes, and (6) writing of each theme for publication. The focus group interview transcript was initially coded by two of the researchers, K.M. and K.P. Codes were deductively analysed (i.e., without trying to fit data into a pre-existing coding frame or theoretical framework). Two members of the research team (S.C. and B.G.) then reviewed preliminary (emerging themes) and data were re-analysed until consensus on the final themes was reached. As themes were determined from a single focus group interview, it is unlikely that saturation of data was achieved in the study. However, the trustworthiness of the findings was enhanced through rich description of the participants and triangulation of researchers involved in data analysis.

Quantitative health data are presented as mean and 95% confidence intervals (95% CI). Data analysis was completed using the software package SPSS (SPSS^®^ Statistics, version 29, SPSS Inc., Chicago, IL, USA). Percentage (%) change, mean differences (95% CI) and Cohens d were calculated to express the magnitude of change with values of <0.2, 0.2–0.59, 0.6–1.19, 1.2–1.99 and ≥2.0 defined as trivial, small, moderate, large or very large respectively [26].

## 3. Results

The 13 recruited participants completed all training sessions throughout the 6-week walking football intervention, corresponding to a 100% attendance.

### 3.1. Focus Group Interview

Three themes that captured the participants’ experiences were identified: (1) “Opportunity to jump back in with what we can physically do”, (2) Social connection and feeling “part of something bigger”, and (3) “It’s how our brain is engaged”. These themes are described below. Pseudonyms are used to protect the anonymity of individual participants.

#### 3.1.1. “Opportunity to Jump Back in with What We Can Physically Do”

Seven of the participants (Robert, William, David, Stephen, Sarah, Margaret and Kathleen) reflected on their increased confidence in their physical abilities following participation in walking football. Specifically, participants identified benefits with strength, balance, flexibility, and endurance. Kathleen described, “it’s amazing how much I’ve noticed my physical fitness has improved. And also, a sense of confidence, of just moving around in a sports situation, that normally I would be terrified.” Similarly, David described, “It’s been a real wake-up call for me. The flexibility and balance aspects that I had lost, but were still able to be recovered, I wouldn’t have realised that.” Increased confidence in physical abilities enabled participation in other activities outside of walking football, such as other sporting groups, supermarket shopping, and travel. Factors that supported the opportunity to “jump back in with what we can physically do” included indoor play with good lighting, flat floor surfaces and temperature control, as well as use of a trained exercise scientist to facilitate stretching/warm-up activities. Participants appreciated evening sessions and adaptations to the game rules to ensure their safety. Flexibility in attendance allowed for continued participation in the event of illness or injury. Participants were, however, concerned about inconsistency in rules when competing against other teams and two participants described being afraid of some of the male players from other teams who were “more physical and aggressive”.

#### 3.1.2. Social Connection and Feeling “Part of Something Bigger”

Nine of the participants described how participation in walking football had led to friendships and a sense of social connectedness. As Kevin described:

“We’ve always focussed on the fun and the fitness and the inclusiveness. So, that’s really shone through, whereas I know there were other clubs that they’re really serious about it and that’s okay. But I think having that be an objective of what the purpose is, is really important”.

Lisa further described, “we’ve got some really lovely friends in there, and it’s really lovely to associate with them every week. We just look forward to coming each week, just for the fitness and meeting people and all.” One participant had not played team sports before, describing the excitement of team comradery and interactions during competitions. Three of the participants described the feeling of team comradery as being “part of something bigger”. One couple (Margaret and Robert) planned to travel internationally to connect with other walking football groups. Hannah and Grace described feeling more connected to their children and grandchildren as they now had a shared interest in football, and Sarah and Jack described a new interest in watching national football games online.

##### 3.1.3. “It’s How Our Brain Is Engaged”

Five participants (William, David, Kevin, Robert, and Grace) described cognitive and psychological benefits from participating in walking football. For example, William described,

“I think the mental side of it, too. Like trying to be in some ways, tactical about, trying to figure out where you’re going to go, where the ball’s going to go, keeping your mind ticking over and thinking about, where am I going to go, who can I pass to, where can I position myself? So, yes, really thinking about that side of it too, apart from the physical side, I think that mental stimulation and activity is a big part of it”.

Similarly, David described, “Strategising about the game keeps my mind active and engaged.” Participants also described positive affective states when playing walking football. David described feelings of “excitement” when engaging in competitions with other teams. Robert described “dreaming” about competing internationally. Grace and Kevin described experiencing “enjoyment” and “fun” when socialising with other players. The competitive nature of the game contributed to participant’s enjoyment and engagement in walking football, with William describing his determination to continue playing. Increased physical abilities further contributed to feelings of self-efficacy and self-confidence.

### 3.2. Anthropometry and Functional Outcomes

Anthropometrical and functional outcomes are presented in Table 1. Trivial effects were observed for body mass, BMI and both the CSRT and BST. Small-to-large magnitude changes in functional outcomes were observed through increases of 13.2% walking distance in the 6MWT, and 10.1%, 10.9% and 13.4% increases in repetitions for the CST, ACT and 2MST respectively. There was also a moderate magnitude reduction of 14% in time to complete the TUG.

## 4. Discussion

Participant experiences included perceived improved strength, balance, flexibility, and endurance, along with perceived increased self-efficacy, and increased participation in other life activities. These outcomes potentially contribute to enhanced well-being and quality of life, reflected through identified opportunities for friendship, social connection, cognitive stimulation, and positive emotional/psychological experiences. Flexibility and timing of sessions was identified as important to promote participation. The participants’ perceived experiences of improved physical capacity were reflected by small-to-large magnitude increases in several functional health outcomes suggestive of improved walking capacity, agility and upper and lower body muscular strength following the 6-week walking football program.

Participants in this program perceived that walking football contributed to social connectedness and “feeling part of something bigger”, which aligns with previous findings on the experiences influencing walking football initiation, maintenance and sustainability in older adults [12,15]. These findings are important given that rural older adults are at an increased risk of experiencing loneliness and social isolation than their urban counterparts [27]. Various factors can contribute to loneliness and social isolation in older rural populations, including health and mobility issues, lack of adequate infrastructure and rural cultural power dynamics [28]. In the present study, participants described the importance of an accessible indoor playing environment, flexibility in attendance, and an inclusive playing culture for enabling their participation, which has been described by Romein et al. [29] as necessary for the delivery of an adaptive sport program. While an indoor environment was possible in this setting, it is not always available in rural areas, which makes flexibility and inclusiveness even more important when setting up adaptive sports programs. Participants attributed the accessible and inclusive group setup to feelings of social connectedness and other psychological benefits, including positive affective experiences such as enjoyment and fun. The importance of group characteristics and culture has been identified in shaping psychological well-being and resilience in rural adults [30]. It is therefore recommended that rural adaptive walking sports programs incorporate similar inclusive and accessible design considerations to promote optimal health, well-being, and social connection among older adults.

Considering that cognitive decline is associated with a range of functional impairments in older adults, the finding that participation in walking football had perceived cognitive and psychological benefits highlights the potential that adaptive walking sports have in promoting health and functional independence in older adults. Furthermore, the relationship between values and perceptions of health and physical activity has previously been identified as an important driver of interest in and commitment to walking football groups in older adults [14,18]. Participants in the present study highlighted the importance of cognitive stimulation and challenge in maintaining their executive functioning skills. Positive affective states, including excitement, fun, and enjoyment experienced during play may contribute to older adult’s intrinsic motivation and general mental health and well-being, and this should be investigated in future research.

Despite the low sample size and short intervention duration, the small-to-large magnitude improvements in functional health outcomes observed in this pilot study are in alignment with previous findings from non-controlled walking football programs [16,31]. Furthermore, the improved health outcomes demonstrated in this short duration, low dose intervention, are similar to previously reported findings in higher dose case-controlled interventions conducted in metropolitan settings [15] including those implemented for people with cancer [17] and hypertension [32]. Whilst it is encouraging to determine that such benefits can be attained with only six hours of organized adaptive sport, this falls below the Australian Institute of Health and Welfare physical activity recommendations for older populations (≥65-years) of completing 30 min of moderate intensity physical activity on most, preferably all, days [33] and may limit further health benefits, as it appears the lower exercise dose was insufficient to maintain body mass, in line with previous findings [34]. Consequently, further investigation into the value of suitably powered, lower dose, case-controlled interventions is needed to confirm these initial novel findings, with gold-standard measures of body composition also warranted in future research.

### Limitations and Further Research

As this was a pilot study, the primary limitations reside in the non-controlled, observational study design and short duration of the intervention limiting the ability to determine effect beyond doubt. However, the addition of qualitative findings that support the quantitative outcomes partially overcomes this limitation. Furthermore, due to these limitations, no comparisons were attempted between genders, and in the future, suitably powered case-controlled interventions on the effects of walking football in rural older populations that explore differences between genders, are warranted, given the potential benefits of this form of physical activity. It is also important to acknowledge that the Senior Fitness Test Battery only provides an indication of physical function and some of the initial functional health outcomes reported would be considered low [35], which may have contributed to the moderate-to-large magnitude improvements in performance. Such improvements might not be replicated in older populations with a higher initial baseline level of fitness or where gold-standard measures of physical function are used. Additionally, whilst participants were asked to maintain their typical physical activity behaviors throughout the duration of the study, this was not monitored or controlled. Consequently, it is possible that changes in physical activity outside of the intervention contributed towards the changes in functional health outcomes observed in this study; however, whilst this should be considered a positive outcome if it were true, the implementation of validated methods to assess physical activity levels is recommended in future research.

Finally, various strategies were used to enhance the trustworthiness of the qualitative (focus group) findings including provision of rich descriptions of the study context and participants, and triangulation of researchers for data analysis, although findings may have been impacted by volunteer bias (leading to positive study outcomes) and use of a single study site. To further enhance the reliability and generalizability of the qualitative results, smaller-sized focus groups and/or individualised interviews alongside the use of standardised measures of psychological and psychosocial health are recommended in future research.

## 5. Conclusions

This pilot study provides initial evidence that 6-weeks of walking football promotes positive experiences, including increased confidence in physical abilities, social connectedness and support, and perceived cognitive and psychological benefits in a rural population of older adults. Quantifiable small-to-large magnitude improvements were observed in assessments of walking capacity, agility and upper and lower body muscular strength, and the data should be used to power appropriately controlled studies that also explore differences between genders to determine the efficacy of walking football to improve the health and well-being of rural older adults.

## Figures and Tables

**Table 1 sports-13-00272-t001:** Mean (95% CI) for anthropometrical and functional outcomes before and after a 6-week walking football intervention.

	Pre			Post					
	Total	Male	Female	Total	Male	Female	% Change	Mean Difference	Cohens d
Mass (kg)	83.9 (69.7–90.2)	82.0 (67.6–98.8)	77.9 (66.5–87.4)	85.4 (71.4–92.1)	83.9 (69.3–101.3)	79.5 (68.3–88.3)	1.7	1.5 (−2.5–−0.4)	0.07
BMI (kg/m^2^)	28.2 (23.5–30.3)	25.7 (21.8–30.8)	28.1 (23.9–32.2)	28.8 (24.1–30.8)	26.2 (21.9–31.4)	28.7 (24.7–32.5)	1.8	0.5 (−0.9–−0.2)	0.08
CST (reps)	13.9 (12.7–15.4)	13.1 (11.2–15.0)	15.0 (14.0–16.0)	15.2 (14.3–16.3)	14.7 (13.6–16.1)	16.0 (15.3–17.0)	10.1	1.3 (−2.3–−0.3)	0.49
ACT (reps)	19.0 (16.1–22.5)	16.3 (14.5–18.5)	22.2 (18.0–26.3)	21.4 (18.6–24.4)	19.0 (16.8–21.7)	24.0 (21.4–27.8)	10.9	2.4 (−3.9–−0.8)	0.49
TUG (s)	5.7 (5.1–6.1)	5.8 (5.1–6.5)	5.3 (4.8–5.8)	4.9 (4.4–5.2)	4.9 (4.2–5.5)	4.8 (4.5–4.9)	14.0	0.8 (0.5–1.2)	−0.9
CSRT (cm)	Right: 1.8 (−1.56–8.9)	1.7 (−4.1–6.0)	5.7 (−2.0–13.3)	Right: 2.1 (−1.4–7.6)	0.1 (−5.7–5.1)	6.3 (1.5–11.3)	16.7	0.3 (−5.8–5.3)	0.02
	Left: 1.8 (−1.1–8.9)	2.3 (−4.7–7.7)	5.5 (−0.6–11.5)	Left: 3.1 (−1.3–8.8)	0.3 (−6.0–5.9)	7.2 (1.8–13.1)	72.2	1.3 (−6.5–3.9)	0.12
BST (cm)	Right: −16.4 (−21.5–−8.2)	−15.3 (−25.8–−5.6)	−14.3 (−21.2–−8.0)	Right: −14.1 (−18.7–−5.3)	−15.7 (−24.9–−6.1)	−8.3 (−15.9–−2.2)	13.4	2.23 (−5.6–1.2)	0.19
	Left: −19.6 (−24.1–−8.37)	−20.5 (−30.6–−8.4)	−12.0 (−18.8–−5.2)	Left: −18.4 (−24.9–−10.4)	−22.2 (−29.7–−11.0)	−13.1 (−20.2–−7.3)	6.5	1.19 (−3.2–5.6)	0.08
2MST (reps)	196.2 (181.1–216.2)	192.3 (168.6–217.8)	205.0 (188.0–222.6)	217.7 (198.9–238.8)	213.3 (187.8–232.0)	224.3 (198.2–249.7)	10.9	21.5 (−39.2–−3.8)	0.72
6MWT (m)	538.3 (510.6–566.1)	546.7 (511.7–595.8)	530.0 (503.8–550.7)	609.3 (571.8–646.8)	608.3 (571.3–649.9)	610.3 (549.7–663.3)	13.2	71 (−97.3–−44.7)	1.6

Abbreviation ACT = 30-s arm curl test; BMI = Body Mass Index; BST = back scratch test; CSRT = chair sit and reach test; CST = 30-s chair stand test; TUG = 8-foot timed up and go test; 2MST = 2-min step in place test; 6MWT = 6-min walk test.

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
