# Peer review of "Experiences and Functional Health Outcomes Associated with a Walking Football Program in Rural Older Adults: A Pilot Study"

_sports, 2025, doi:10.3390/sports13080272_

Round 1

Reviewer 1 Report (Previous Reviewer 1)

Comments and Suggestions for Authors

The authors have addressed the comments, and the manuscript has been improved for its potential publication. Thank the authors for their works on this manuscript.

Author Response

Please refer to the attached response. 

Reviewer 2 Report (Previous Reviewer 2)

Comments and Suggestions for Authors

Thank you for the opportunity to review the article again.

I greatly appreciate the changes made by the authors, which I believe have significantly improved the concept conveyed by the paper.

I have only one report:

  • In Table 1, I invite authors to check the Mean difference values, which are all expressed as negative variations, even in cases where the values increase.

Congratulations to the authors for their work.

Author Response

Please refer to the attached response. 

Reviewer 3 Report (Previous Reviewer 3)

Comments and Suggestions for Authors

The new version of the manuscript addressed well most of the comments and suggestions from the first review. However, I still think it is relevant to investigate if there was an effect of sex. It would be beneficial including two more tables similar to the table 1, but presenting data of male and female individuals separately. 

Author Response

Please refer to the attached response. 

This manuscript is a resubmission of an earlier submission. The following is a list of the peer review reports and author responses from that submission.

Round 1

Reviewer 1 Report

Comments and Suggestions for Authors

The study targets a largely under-researched population—rural older adults—using a relatively novel intervention (walking football), thereby filling a meaningful gap in the literature. Plus, the combination of pre-post quantitative measures and qualitative focus group data allows for a comprehensive exploration of both functional outcomes and participant experiences.

However, the authors need to address the following comments for its publication.

The introduction tends to list prior studies without synthesis or critique. Although the authors mentioned the limitations of the previous studies in lines 57 to 61 and this study is the first study to include both genders residing in rural locations, there is little effort to problematize the literature or to show how this study provides a distinctive practical and conceptual contribution. The authors need to strengthen the need of the study by persuasively presenting how gender differences and different residing areas can play an important role in differentiating the experience and physiological and psychological benefits of participating in walking football.

The absence of any control group renders it impossible to attribute observed improvements to the intervention itself. The authors addressed this limitation in the discussion, but if possible add in the discussion some previous studies showing controlling effects of walking football.

The manuscript fails to engage meaningfully with theories of ageing, social cognitive theory, or health behavior frameworks. The authors are required to formulate a conceptual model that establishes a link between walking football and the psychosocial outcomes of the study.

This study employs a mixed methods approach; consequently, further information regarding the participants is required.

Thematic analysis is described in brief, but no discussion is provided of coding frameworks, intercoder reliability, or data saturation procedures.

Themes reported are uniformly positive. Were any dissenting or negative views expressed but omitted? This raises concern about selective reporting.

The effect sizes of Cohen's findings are negligible or of a trivial nature. It is imperative that the authors address these findings in the discussion section.

Author Response

Thank you very much for taking the time to review this manuscript. Please find the detailed responses and the corresponding revisions/corrections highlighted in the attached reviewer report and revised manuscript respectively. 

Reviewer 2 Report

Comments and Suggestions for Authors

Thank you for the opportunity to review this article. The authors' aim was to evaluate the effects of a 6-week walking football programme on psychological, functional and anthropometric parameters. The paper is written in a simple but clear way. Unfortunately, there are few quantitative measures to evaluate the impact of the intervention.

Here are my comments:

  • Abstract, line 13: I suggest replacing “body mass” with the more generic “body composition”.
  • Abstract, line 13: I suggest replacing “assessments of functional fitness” with “functional assessments”
  • Material and Methods, line 72: I invite the authors to indicate which university ethics committee approval was obtained.
  • Material and Methods, lines 90-93: I do not think it is necessary to specify the procedure for measuring such well-known parameters. I therefore suggest that the authors remove it from the text.
  • Material and Methods, lines 99-101: Why did you allow people to try the measurements twice? I believe that in some cases there is a risk of a learning effect that could have distorted the assessment.
  • Material and Methods, line 145: I encourage authors to remove the "structured... structure" repetition
  • Results, Table 1: Given the baseline level of the 6MWT (538.3 metres), I find an improvement of over 13% surprising, given that the proposed dose of physical activity was low (1 hour/week). Although the authors state (lines 314-316) that there may have been an increase in spontaneous physical activity, I ask how they interpret this finding. 
  • In general, I think it would have been useful to include an instrument to assess the impact from a psychological point of view in a quantitative way, rather than just interviews.

Author Response

(The authors gave the same response as above.)

Reviewer 3 Report

Comments and Suggestions for Authors

The paper entitled “Psychosocial and Functional Benefits of Walking Football in Rural Older Adults” has evaluated phycological and functional changes induced by 6 weeks of walking football practice. Thirteen individuals were submitted to functional physical testes pre and post the intervention as well as a focus group interview. Overall, the introduction is well written, and methods has a good explanation of the study design, but a few adjust can be done. The results are clearly presented, but I believe that it would be interesting to evaluate the response between sexes as well. In my opinion the discussion section is quite incomplete. Although many functional testes were carried out, the outcomes were not explored in the discussion, as well as the demographic data. For example, it was observed an increase in body mass and BMI, which was not explored by the authors. Please, see below some comments/suggestions:

Line 31: In “This demographic shift increases the risk of developing chronic, ...” Is it an increase of the risk or increase in the prevalence? It is not clear how demographic changes increase the risk of developing chronic diseases, please provide more context and the rationale.

Lines 62-65: I suggest re-phrase the whole paragraph. Mainly, removing the “benefits of participating in walking football” and replace by “effects of participating in walking football”. It is more appropriate to state that the aim is to investigate the effects of the intervention, and your hypothesis is that it is beneficial.

Line 72: Please indicate the ID of ethical approval.

Line 76: Minor suggestion, use “Thirteen” at the beginning of the phrase.

Line 78: Is 13 a subset of the participants? Please inform how many people are part of the group. Is this sample size representative for the entire group population?

Line 105: Minor suggestion, I think “time in seconds” sounds better than “number of seconds”

Line 118: Typo: 2MST instead 2ST.

Lines 187-188: Usually, we don't report the name of the participants in the research articles. Have the participants agreed with sharing their names? Is it necessary to report it? I suggest using ID.

Section “3.2. Anthropometry and Functional Outcomes”: I suggest including analysis of male and female separated. It would be interesting to evaluate the effect of sex.

Table 1: Include in the legend the meaning of each abbreviation.

Discussion section: The functional outcomes where not discussed as well as demographic data, for example, why the body mass increased? Please explore more these findings.

Author Response

(The authors gave the same response as above.)

Round 2

Reviewer 1 Report

Comments and Suggestions for Authors

The authors have successfully addressed the comments, and the manuscript has improved for its publication. Thanks for their efforts on the manuscript.

Reviewer 2 Report

Comments and Suggestions for Authors

Thank you for the opportunity to review the article again. I would like to thank the authors for their efforts in improving the quality of the article. However, I think there are still some critical points, as explained in the comments.

Here are some comments on the authors' answers::

  • Material and Methods, lines 116-117: Why did you allow people to try the measurements twice? The authors state in the text that the measurement procedure was performed twice for each test (excluding 2MST and 6MWT). However, as they replied to my previous comment, since the original method already requires the measurement to be repeated twice (and I believe that the authors actually followed this protocol), I consider that this formulation may confuse the reader.
  • Results, Table 1: Given the baseline level of the 6MWT (538.3 metres), I find an improvement of over 13% surprising, given that the proposed dose of physical activity was low (1 hour/week). I agree with the authors that, given the starting level, a 13% increase in 6 weeks would not be impossible in absolute terms. However, it remains in my opinion a surprising finding considering the small amount of activity that is proposed in the study (a total of 6 hours of unstructured activity, without monitoring the intensity at which it was carried out). Unfortunately, questionnaires were not submitted at baseline or at the end of the intervention to assess physical activity levels, which would have provided useful data.

Here the general comment on the article:

  • From the title of the paper, 'Psychosocial and functional benefits of walking football in rural older adults', one might expect a study measuring the effects of walking football on the chosen outcomes. However, the methodology chosen by the authors appears to prioritise qualitative interviews that investigate participants' perceptions of the effects of walking football. This approach does not evaluate objective aspects or submit any questionnaires or psychosocial information to the baseline. Conversely, the evaluation of the impact of the proposed physical activity on various functional aspects is pushed to the background, despite it appearing to have been carried out with greater precision (although there is a lack of data to help interpret the results).
  • While I appreciate the authors' work, I believe that, in its current state, the article has several issues and does not align with what the authors state in the abstract ('This study investigated the experiences of walking football among older adults in rural areas and its associated functional health and well-being outcomes').
  • I would suggest placing greater emphasis on the functional data obtained, and using the psychosocial results as a starting point for future research. This could then provide a basis for collecting objective data.